# Diagnostic Biomarker Candidates Proposed Using Targeted Lipid Metabolomics Analysis of the Plasma of Patients with PDAC

**DOI:** 10.3390/cancers17182988

**Published:** 2025-09-12

**Authors:** Sung-Sik Han, Sang Myung Woo, Jun Hwa Lee, Joon Hee Kang, Sang-Jae Park, Woo Jin Lee, Hyeong Min Park, Jung Won Chun, Su Jung Kim, Hyun Ju Yoo, Kyung-Hee Kim, Soo-Youl Kim

**Affiliations:** 1Center for Liver and Pancreatobiliary Cancer, National Cancer Center, Goyang 10408, Republic of Korea; sshan@ncc.re.kr (S.-S.H.); wsm@ncc.re.kr (S.M.W.); spark@ncc.re.kr (S.-J.P.); lwj@ncc.re.kr (W.J.L.); parkhmgs@ncc.re.kr (H.M.P.); deli4927@ncc.re.kr (J.W.C.); 2Research Institute, National Cancer Center, Goyang 10408, Republic of Korea; junhwalee@ncc.re.kr; 3Department of Cancer Biomedical Science, National Cancer Center Graduate School of Cancer Science and Policy, Goyang 10408, Republic of Korea; wnsl2820@gmail.com; 4Department of Convergence Medicine, Department of Digital Medicine, Asan Institute for Life Sciences, Asan Medical Center, University of Ulsan College of Medicine, Seoul 05505, Republic of Korea; sujung1008@amc.seoul.kr (S.J.K.); yoohyunju@amc.seoul.kr (H.J.Y.); 5Proteomics Core Facility, Research Core Center, Research Institute, National Cancer Center, Goyang 10408, Republic of Korea; kyungheekim@kookmin.ac.kr; 6Biopharmaceutical Chemistry Major, Department of Applied Chemistry, School of Science and Technology, Kookmin University, Seoul 02707, Republic of Korea

**Keywords:** pancreatic cancer, diagnostic biomarker, lipid metabolomics, CA19-9

## Abstract

We recently discovered that tumors rely on blood fatty acids as an energy source for growth. Therefore, we investigated biomarkers in the lipid fractions of plasma from patients with pancreatic ductal adenocarcinoma (PDAC) for the screening diagnosis of PDAC. We screened common fatty acid types in human and mouse plasma samples using a non-targeted approach. Subsequently, we identified targets in human plasma that could distinguish between healthy individuals and patients with cancer. Based on an average AUC for LR models with 11 or more phospholipids, the separation performance between healthy individuals and patients with cancer was 0.9207, while the addition of CA19-9 to the LR models resulted in a separation performance of 0.9427 for the validation set. We identified candidate metabolites and algorithms that could serve as biomarkers in the lipid fraction of serum from patients with pancreatic cancer. However, more validation sets and multicenter analyses are needed to determine clinically practical implications.

## 1. Introduction

Pancreatic ductal adenocarcinoma (PDAC) accounts for over 90% of all pancreatic cancers and remains one of the deadliest cancers globally, with a 5-year overall survival rate of 10% [1,2,3]. This dismal prognosis is mainly because PDAC is usually diagnosed at an advanced stage. Early-stage PDAC is often asymptomatic or presents with nonspecific symptoms, so over 80% of patients are unresectable at diagnosis. As a result, the median survival for individuals with advanced-stage PDAC is measured in months, and by 2030, PDAC is expected to rank second among cancer-related causes of death in the United States. Although surgical resection of stage I and II PDAC can increase the 5-year survival rate to nearly 44%, fewer than 20% of patients qualify for resection at the time of diagnosis. Therefore, developing reliable, non-invasive biomarkers for early detection is crucial.

Currently, the only FDA-approved serum biomarker for PDAC diagnosis and monitoring is carbohydrate antigen 19-9 (CA19-9) [4,5]. In symptomatic patients, CA19-9 achieves a sensitivity of 79% and a specificity of 80%. However, its performance in early or asymptomatic PDAC is poor, with sensitivity dropping below 20%. Moreover, 5–10% of patients are Lewis antigen-negative and cannot synthesize CA19-9, resulting in false-negative test results [5,6]. In addition, elevated CA19-9 levels may be observed in benign conditions, such as cholangitis, pancreatitis, and poorly controlled diabetes, leading to false-positive results. As a result, CA19-9 is primarily used for monitoring the therapeutic response and disease recurrence, rather than for screening purposes. Other serum markers, such as carcinoembryonic antigen (CEA) and cancer antigen 125 (CA-125), have been evaluated in combination with CA19-9. While CEA and CA-125 individually exhibit a sensitivity of 51%, a panel combining these markers can improve sensitivity to about 74%, though with reduced specificity, since these antigens may also be elevated in benign hepatic or gastrointestinal disorders and other malignancies [3]. Emerging liquid biopsy approaches, including circulating tumor DNA (ctDNA), circulating tumor cells (CTCs), and exosomal proteins, have shown promise. However, none has yet demonstrated consistent sensitivity and specificity for early PDAC detection in large-scale prospective cohorts [7]. Overall, existing blood-based biomarkers lack the accuracy required for reliable early-stage PDAC screening, underscoring an urgent need to identify novel, non-invasive biomarker candidates.

One promising approach is to extract and analyze fatty acids from patients’ blood samples. PDAC cells undergo extensive lipid-metabolic reprogramming to support malignant proliferation and survival. Specifically, these cells upregulate both de novo lipogenesis (via enzymes such as fatty acid synthase (FASN) and stearoyl-CoA desaturase (SCD)) and lipid uptake, resulting in altered concentrations of free fatty acids in the circulation. Under nutrient deprivation or hypoxia conditions, PDAC cells shift toward fatty acid oxidation (FAO) to generate ATP and NADPH, promoting their survival under metabolic stress. Such metabolic switching results in detectable changes in plasma free fatty acid and acylcarnitine profiles. Furthermore, rapidly dividing PDAC cells require large quantities of membrane phospholipids (e.g., phosphatidylcholine and phosphatidylethanolamine), which are derived from fatty acid precursors. Consequently, the synthesis and remodeling of specific fatty acid species are reflected in the plasma lipidome. In addition, bioactive lipid mediators, such as lysophosphatidic acid (LPA), sphingosine-1-phosphate (S1P), and prostaglandin E2, originate from fatty acid metabolic pathways and play critical roles in tumor proliferation, invasion, and immune modulation. Fluctuations in these lipid signaling molecules can serve as indirect indicators of PDAC presence and progression. Moreover, PDAC tumors often upregulate cholesterol synthesis and uptake to maintain membrane fluidity and to support receptor-mediated signal transduction; in this context, fatty acid–derived precursors (e.g., acetyl-CoA) feed into altered cholesterol homeostasis, resulting in plasma fatty acid profiles that differ significantly from those of healthy individuals. Fatty acids are relatively stable in plasma and can be accurately quantified using established techniques, such as gas chromatography–mass spectrometry (GC-MS) and liquid chromatography–mass spectrometry (LC-MS); therefore, a lipidomics-based approach targeting fatty acid composition offers enhanced sensitivity and specificity compared to traditional protein markers. Cancer cells can use fat or fatty acids in the blood as an energy source. The reason is that there are 50~200 µM for acetate [8], free fatty acids (100 to 400 µM) [9], and cholesterol (5.2~6.1 mM), LDL (2.6~3.3 mM), HDL (1.0~1.5 mM), and triglycerides (1.7~2.2 mM) in the human blood [10]. Therefore, considering the amounts of various fuel sources for ATP production, including lipids and free fatty acids, cancer cells depend on FAO for ATP production. Furthermore, a recent review has summarized the mechanism by which these lipids and fatty acids are absorbed by cancer cells through various pathways and used in energy metabolism through fatty acid oxidation [11]. Therefore, in this study, we propose that fatty acids represent a novel class of plasma biomarkers for early PDAC detection.

## 2. Materials and Methods

### 2.1. Non-Targeted Metabolomics Analysis of Human and Mouse Plasma and Selection of a Target Metabolome

A total of 202 plasma samples from healthy subjects (NM; *n* = 99) and patients with pancreatic cancer (PC; *n* = 103) were obtained from the National Cancer Center (Institutional Review Board approval number: NCC2017-0060). A total of 29 plasma samples were also collected from normal mice (NM; *n* = 7) and KPC mice (KPC; *n* = 22). This study was reviewed and approved by the Institutional Animal Care and Use Committee (IACUC) of the National Cancer Center Research Institute (NCCRI; protocols: NCC-21-574B-003, NCC-20-588), which is accredited by the Association for Assessment and Accreditation of Laboratory Animal Care International (AAALAC International) and adheres to the Institute of Laboratory Animal Resources guidelines.

Non-targeted metabolome profiling of human and mouse plasma samples was independently performed. Pooled plasma samples were used for quality control (QC), and the same number of samples from each group was allocated across batches, with two QC samples in each batch. The analysis order was randomized within each batch. Non-targeted metabolomics analysis was performed using LC-MS/MS (Ultimate 3000-LTQ-Orbitrap XL; Thermo Fisher Scientific, Waltham, MA, USA). Metabolomic features, including the mass value (mass-to-charge ratio (*m*/*z*)) and retention time (min), were extracted using Compound Discoverer 3.3 (Thermo Fisher Scientific, Waltham, MA, USA). Metabolites were identified through database searches based on accurate masses with a 10 ppm mass tolerance, isotopic abundance, and MS/MS library matching. Metabolomic features with an annotation confidence level of 2–4 were used for statistical analysis [12]. A metabolomic feature table was created for each human and mouse plasma sample. The following metabolomic features were determined: accurate mass (*m*/*z*), retention time, and normalized peak area. The peak areas in the table were converted to common logarithms. The discriminative performance of each individual metabolomic feature was evaluated in the table as follows: A receiver operating characteristic (ROC) curve was plotted, and the area under the ROC curve (AUC) was calculated and designated as the discriminative performance of the metabolomic feature. Standard discriminative metabolomic features from both human and mouse plasma samples were identified, and MS/MS spectra of the features were obtained if they were not available from the previous non-targeted metabolome profiling. Metabolites with putative identification from MS/MS library searches were selected for further validation using targeted metabolomics platforms. In this study, the selected target metabolites belonged to different lipid categories. Each targeted lipidomics platform comprises lipids in the same subclass, where the lipids share a common chemically functional backbone that influences their biological activities. When targeted lipidomics analysis was performed, other measurable lipids in each platform were also quantified (Figure 1).

### 2.2. Targeted Lipidomics of Human Plasma

Targeted lipidomics analysis was performed on 173 human plasma samples (70 NM and 103 PC samples available) using LC-MS/MS equipped with Agilent 1290 (Agilent, Santa Clara, CA, USA)/Qtrap 5500 (ABSciex, Framingham, MA, USA) (Figure 1). Four targeted lipidomics platforms were used, each consisting of one lipid subclass: acylcarnitines, phospholipids, fatty acid amides, and sphingolipids. Details of the quantitation method for the target lipidome are provided in Appendix A. Data analysis of the lipids was performed using Analyst 1.7.1 (ABSciex). The extracted ion chromatogram (EIC) corresponding to each lipid was used for quantitation. The area under the curve for each EIC was normalized to that of the corresponding internal standard (IS). Since some lipid concentrations for 1 PC sample were not available in the phospholipids platform and CA19-9 was not available for 2 NM samples, lipid concentration data of the remaining 170 (68 NM and 102 PC) samples were used in the subsequent analysis. Similar to the non-targeted metabolomics data, an ROC curve was plotted and the AUC calculated for each lipid to assess its discriminative performance.

The same targeted lipidomics analysis was performed on 180 new human plasma samples (100 NM and 80 PC samples). Since CA19-9 was not available for 4 NM samples and 2 PC samples, measurements from the remaining 174 (96 NM and 78 PC) samples were used to determine how many discriminative lipids from the 170 samples (set A) could maintain discriminative performance for the independent 174 samples (set B). After identifying the discriminative lipids for both sets A and B, the two sample sets were merged into a total sample set C, which was further divided in a 7:3 ratio into a training set (115 NM and 126 PC samples) and a validation set (49 NM and 54 PC samples). The discriminative lipids and CA19-9 were used to train three statistical models: logistic regression (LR), random forest (RF), and support vector machine (SVM) with a radial basis function (RBF) kernel. The LR, RF, and SVM models were built using R (version 4.5.0) with the caret package. The discriminative performance of the three models was evaluated using the validation set.

## 3. Results

### 3.1. Characteristics of Patients with PDAC

The characteristics of the 180 patients included in the study are summarized in Table 1. Set A comprised 102 patients (47 men and 55 women) with PDAC. Their median age was 68.5 years, and the mean BMI of the entire cohort was 23.3 ± 3.0. A history of smoking was reported in 39.2% (*n* = 40) of the patients, and 70.6% (*n* = 72) had diabetes mellitus. In addition, stages I, II, III, and IV accounted for 82.4% (*n* = 84), 5.9% (*n* = 6), 5.9% (*n* = 6), and 5.9% (*n* = 6) of the patients, respectively. The tumor location was the body in 52% (*n* = 53), the head and neck in 43.1% (*n* = 48), the body and head in 1.0% (*n* = 1), and Uncinate process in 3.9% (*n* = 4) of the patients.

Set B comprised 78 patients (44 men and 34 women) with PDAC. Their median age was 64 years, and the mean BMI of the entire cohort was 23.7 ± 3.1. A history of smoking was reported in 48.7% (*n* = 38) of the patients, and 62.8% (*n* = 49) had diabetes mellitus. In addition, stages I, II, III, and IV accounted for 69.2% (*n* = 54), 11.5% (*n* = 9), 12.8% (*n* = 10), and 6.4% (*n* = 5) of the patients, respectively. The tumor location was the body in 37.2% (*n* = 29), the head and neck in 57.7% (*n* = 45), the AoV in 1.3% (*n* = 1), and not available (N/A) in 3.8% (*n* = 3) of the patients.

### 3.2. Non-Targeted Metabolomics Analysis of Human and Mouse Plasma

We detected 684 and 781 metabolomic features in positive and negative modes, respectively, in human plasma samples and 454 and 853 metabolomic features in positive and negative modes, respectively, in mouse plasma samples. The discriminative performance of individual metabolomic features was evaluated using their AUC from ROC analysis. In the negative mode, no common metabolomic feature demonstrated good discriminative performance in both human and mouse plasma. Conversely, in the positive mode, eight common standard discriminative metabolomic features were identified (Figure 2A and Table 2). Figure 2A displays the extracted ion chromatogram for each discriminative metabolomic feature in human and mouse plasma. Among them, Figure 2B illustrates the peak areas of human and mouse plasma samples and the corresponding ROC curves of the fifth common metabolomic feature, which was identified as lysoPC(18:3).

### 3.3. Metabolite Identification Process and Selection of Targeted Lipidomics Platforms

Putative metabolite identification was performed for the standard metabolomic features obtained from non-targeted metabolomics analysis. Table 2 presents the metabolites identified, along with their significantly altered metabolomic features in PC human and KPC mouse plasma compared to their respective controls. Figure A1 displays their MS/MS spectra and database search (or in silico fragmentation) results for putative identification. Three lipidomics platforms (acylcarnitines, fatty acid amides, and phospholipids) were established and used to quantify the lipid concentrations of the standard metabolites identified through targeted lipidomics. An explorative sphingolipids platform was also added. The targeted lipidomics analysis performed also quantified other measurable lipids in each platform. A total of 13 acylcarnitines, 13 fatty acid amides, 12 sphingolipids, and 64 phospholipids (Table A1) were measured across the four lipidomics platforms, with 6 of the 64 phospholipids being relatively quantified.

Since the lipid concentrations of one fatty acid amide (Myristoyl EA) and three phospholipids (PC(12:0/12:0), PE(14:0/14:0), and PE(18:2/18:2)) were unavailable for some of the 170 human plasma samples (set A), the lipid concentration data of 13 acylcarnitines, 12 fatty acid amides, 12 sphingolipids, and 61 phospholipids were used in the subsequent analysis. Similarly, since three fatty acid amides (Myristoyl EA, Arachidonoyl EA, and EPA EA) and three phospholipids (PC(12:0/12:0), PE(18:2/18:2), and PE(22:6/22:6)) were undetectable in some of the 174 human plasma samples (set B), the lipid concentration data of 13 acylcarnitines, 10 fatty acid amides, 12 sphingolipids, and 61 phospholipids were used.

### 3.4. Targeted Lipidomics Analysis of Human Plasma

Similar to the non-targeted metabolomics data, to assess the discriminative performance of each lipid in both sample sets A and B, an ROC curve was plotted and the AUC calculated. The same ROC analysis was also conducted for set C (union of sets A and B). The AUCs of sets A, B, and C were compared in order to exclude lipids with clear batch effects when building statistical models. In set A, 32 (of 98) lipids, in set B, 53 (of 96) lipids, and in set C, 29 lipids had an AUC of ≥0.75. A Venn diagram (Figure 3 and Table A2) was created to illustrate the overlap of lipids. The top three AUCs for set A were observed in fatty acid amides, specifically linoleamide, oleamide, and palmitamide. However, these three fatty acid amides did not reach an AUC of ≥0.75 for set B. Four acylcarnitines (C3-, C4-, C8-, and C12-carnitine) showed AUCs of ≥0.75 for both sets A and B, yet their AUCs were <0.75 for set C (Table A2). This discrepancy was due to the difference in the concentration ranges measured in sets A and B. Table 3 shows the AUCs of CA19-9 and 20 lipids (1 acylcarnitine, 1 sphingolipid, and 18 phospholipids); all had AUCs ≥ 0.75 across all three sets. Although CA19-9 achieved an AUC of ≥0.75 only for set A, it was included in the subsequent analysis to assess the impact of adding it to the statistical models.

All 20 lipids showed an overall pattern of PC < NM, while CA19-9 displayed the opposite pattern. Accordingly, all of the lipids were negatively correlated with CA19-9, although the correlations were very weak, with the maximum absolute coefficient being 0.1151 for set C. At a cutoff of 37 U/mL or higher, CA19-9 achieved a sensitivity of 48.89% in set C. Although CA19-9 showed poor sensitivity in the early stage [13], its sensitivity for stage I was higher (51.45%) than that for stages II-IV (40.48%). Since most of the lipids (18/20, 90%) belonged to a single lipidomics platform, only 18 phospholipids were identified as discriminative lipids for separating the PC group from the NM group. These 18 phospholipids were used to train the LR, RF, and SVM models. The entire set C was divided in a 7:3 ratio into a training set (115 NM and 126 PC samples) and a validation set (49 NM and 54 PC samples). The discriminative performance of the 18 phospholipids was evaluated using the three statistical models: LR, RF, and SVM with an RBF kernel. The 18 phospholipids listed in Table 3 were arranged in descending order of their AUCs in set C. First, two phospholipids with the highest AUCs were used to train the three models. Next, one additional phospholipid with the next-highest AUC was added to each model. Subsequently, the three models were trained with one additional phospholipid one step at a time until the final three models, including all the phospholipids, were constructed. Next, this exact procedure was repeated, but this time, CA19-9 was added at each step. Figure 4 illustrates how the AUCs of the LR model changed with the cumulative addition of phospholipids. After 11 phospholipids were added, the AUC plots flattened for both LR models, without or with CA19-9. However, cumulatively adding phospholipids with descending AUCs to each model did not result in an overall descending trend, with the cumulative addition of phospholipids following the flattened trend. Based on the average AUC for the last eight steps with 11 or more phospholipids, the separation performance of the LR model for the validation set improved from 0.9207 to 0.9427 with CA19-9. For the LR model without CA19-9, the average sensitivity, specificity, PPV (positive predictive value), and NPV (negative predictive value) were 90.74%, 86.22%, 87.90%, and 89.42%. They were 90.74%, 88.01%, 89.32%, and 89.61% for the LR model with CA19-9. Although the AUC of CA19-9 was lower than that of the 18th phospholipid, CA19-9’s opposite pattern positively contributed to the separation between PC and NM groups. The LR model of all 18 phospholipids, without or with CA19-9, resulted in five false-negative cases (four of stage I and one of stages II-IV), which does not indicate poor sensitivity in the early stage, since the validation set consisted of 43 stage I (79.63%) and 11 stage II-IV (20.37%) PC samples. The RF and SVM models (Figure A2) yielded a less flattened curve and did not achieve better performance than the LR model for our validation set samples.

## 4. Discussion

We recently found that cancer cells depend entirely on FAO for ATP production [11,14]. Additionally, we demonstrated that inhibiting FAO has significant anticancer effects. Based on this, we hypothesized that analyzing blood lipid fractions could distinguish between healthy individuals and cancer patients. However, because the lipid fraction contains various types of fats, we narrowed our focus by comparing mouse cancer models with human samples and performing targeted analyses on matching lipid groups. Consequently, eight fatty acids were identified as matches, and their IDs are listed in Table 2. Among these, LysoPC(18:3) showed an AUC of 0.728 in humans (Figure 2B right upper panel) and 1.0 in mice (Figure 2B right down panel). Examining the matched lipid groups between cancer patients and the mouse cancer model in Table 2 reveals three categories: acylcarnitine, fatty acid amide, and phospholipid groups. We also added one more group—sphingolipids—for analysis. We performed targeted lipidomics on human sets A/B/C using these groups. When filtering for AUC values above 0.75, 20 fatty acids were identified (Table 3). Excluding two ceramides, all 18 are from the phospholipid group. Using this data, we analyzed the pattern of LR model AUC changes as more phospholipids were added across two analysis sets (Figure 4). The training set was used to build the model, and the validation set was used to evaluate the model’s results. Both sets were selected arbitrarily. Results indicated that after adding 11 phospholipids, the AUC curves of both LR models flattened, regardless of whether CA19-9 was included. The AUC of CA19-9, currently FDA-approved as a biomarker, was 0.7571 in set A, 0.7356 in set B, and 0.7489 in set C. The AUC of the validated model with 11 phospholipids was 0.9207. The combined model achieved a separation performance with an AUC of 0.9427. These findings strongly suggest that measuring specific fatty acids in plasma could effectively determine the progression status of cancer.

Some of the lipids or fatty acids identified as biomarker candidates in this study had been reported previously. Lysophosphatidic acid (LPA), a phospholipid, has been used as a biomarker previously [15,16]; however, it was not available on our phospholipids platform. Previous studies [17,18] have also discussed ceramide, sphingosine-1-phosphate (S1P), S1P/ceramide, or ceramide/S1P; the C18:1 ceramide was considered as a biomarker by Rixe et al. [17], whereas Guillermet-Guibert et al. did not specify a particular ceramide [18]. Seven ceramides, including C18:1, were available; however, S1P was not part of our sphingolipid platform. C18:1 did not meet our criterion of AUC ≥ 0.75 for the measurements we obtained; therefore, it was not used as a biomarker in this study. Wolrab et al. suggested several sphingomyelins, ceramides, phosphatidylcholines (PCs), and one lysophosphatidylcholine (lysoPC) [19]. Specific PCs and lysoPC species can serve as biomarkers for drug-induced lung disease [20], lung cancer [21], and Alzheimer’s disease [22]. Multiple PC species exhibited dysregulated levels in non-small cell lung cancer, particularly increased saturated/monounsaturated forms (e.g., PC(15:0/18:1), PC(18:0/16:0)), and decreased polyunsaturated forms [23]. Reduced levels of specific lysoPCs, such as lysoPC(16:0) and lysoPC(18:0), are associated with lung cancer and can help distinguish early-stage lung cancer from other lung diseases [21]. Among them, lysoPC(18:2) was also identified in our study. Additionally, PC(32:0) and PC(O-38:5) may correspond to PC(16:0/16:0) and PC(P-18:0/20:4) on our platform for phospholipids. Only lysoPC(18:2), with an AUC of ≥0.75, was used as one of the phospholipid biomarkers in this study.

## 5. Conclusions

Recently, we discovered that cancer cells are completely dependent on fatty acids for ATP production [11,14]. Therefore, we analyzed the diagnostic potential of specific fatty acid levels in blood by investigating how they respond to tumor growth through animal experiments and patient blood tests. We identified 18 candidate fatty acid metabolites that could serve as biomarkers in the serum lipid fractions of pancreatic cancer patients, all of which were found to be reduced in these patients. Furthermore, we developed an algorithm utilizing these markers and demonstrated a 25% improvement in discriminatory power compared to the AUC of CA19-9. Combining the AUC of CA19-9 with this algorithm further improved discriminatory power by 2.38%. The high AUC values of the model incorporating the biomarkers identified in this study with CA19-9 suggest that these markers have potential as novel metabolic markers for pancreatic cancer. Therefore, we identified candidate metabolites and algorithms that could serve as biomarkers in the lipid fractions of plasma from patients with pancreatic cancer. More validation sets and multicenter analyses are needed to determine clinically practical implications.

## Figures and Tables

**Figure 1 cancers-17-02988-f001:**
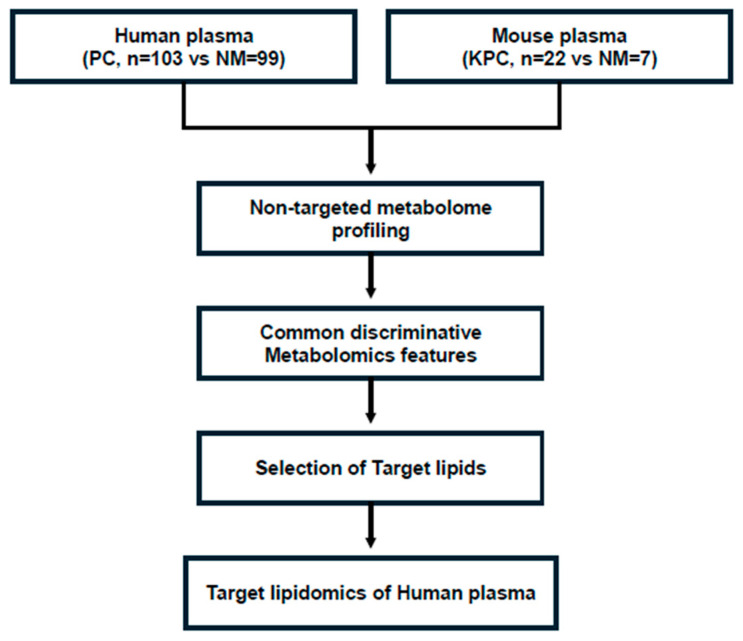
Strategy for deriving candidate lipid metabolites for screening patients with PDAC. Derivation of common target lipid metabolites through non-targeted metabolomics analysis of the plasma of human patients with cancer and a mouse cancer model using human PDAC cells.

**Figure 2 cancers-17-02988-f002:**
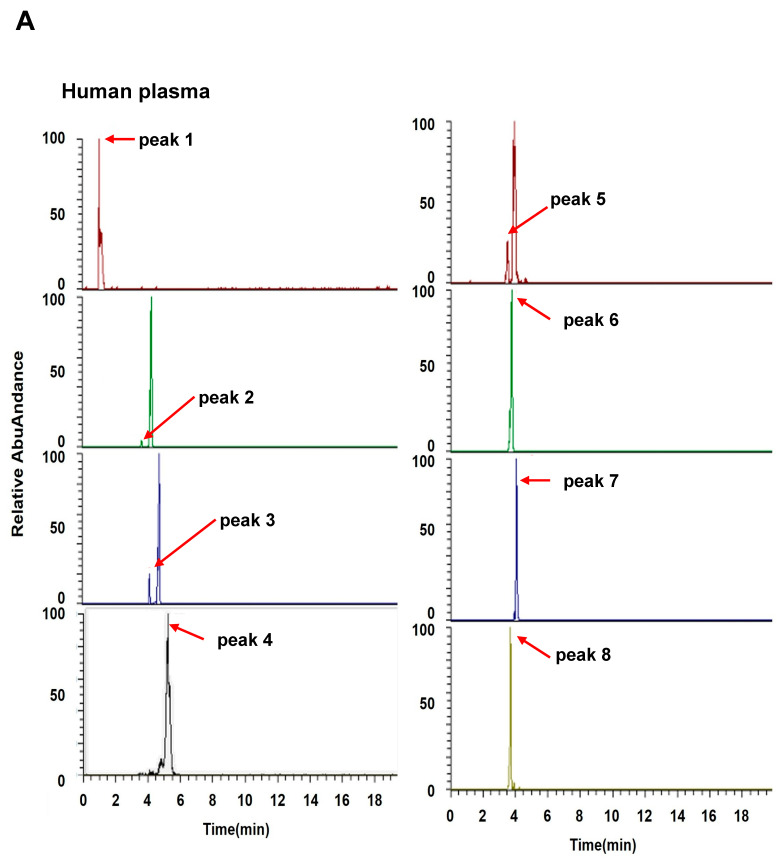
Extracted ion chromatograms and peak areas of human and mouse plasma samples. (**A**) The ion chromatogram revealed distinct metabolomic features in PC human and KPC mouse plasma compared to their matched controls. (**B**) Peak areas of human and mouse plasma samples, along with corresponding ROC curves of the 5th common metabolomic feature: Upper left panel: peak areas of human plasma samples, red dot-pancreatic cancer patients, blue dot-normal persons. Upper right panel: ROC curve of human plasma samples, Lower left panel: peak areas of mouse plasma samples, red dot-KPC mice, blue dot-normal mice, and Lower right panel: ROC curve of mouse plasma samples.

**Figure 3 cancers-17-02988-f003:**
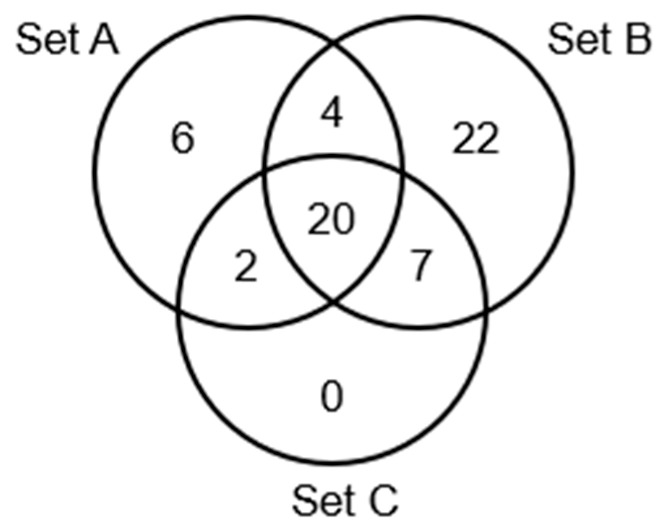
Venn diagram of lipids with AUC ≥ 0.75 in sets A, B, and C.

**Figure 4 cancers-17-02988-f004:**
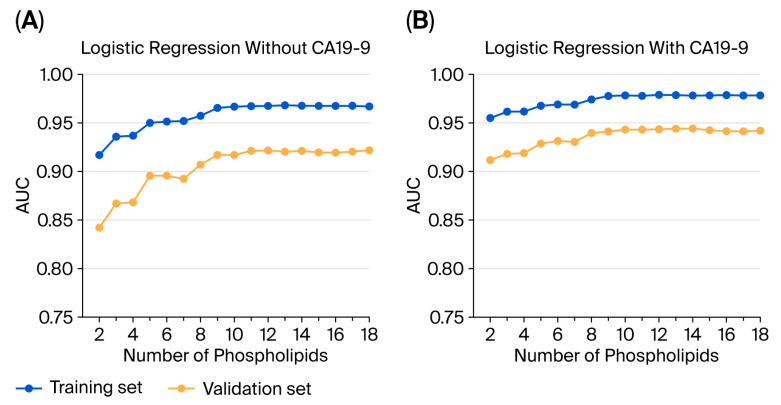
AUC plots of logistic regression models with the cumulative addition of phospholipids arranged in descending order of the AUC (**A**) without CA19-9 and (**B**) with CA19-9.

**Table 1 cancers-17-02988-t001:** Baseline characteristics of patients with pancreatic ductal adenocarcinoma.

Variable	Set A (*n* = 102)	Set B (*n* = 78)
Sex, *n* (%)		
Male	47 (46.1%)	44 (56.4%)
Female	55 (53.9%)	34 (43.6%)
Median age, years	68.5	64
BMI, mean ± SD	23.3 ± 3.0	23.7 ± 3.1
Smoking history, *n* (%)	40 (39.2%)	38 (48.7%)
Diabetes mellitus, *n* (%)	72 (70.6%)	49 (62.8%)
Tumor stage, *n* (%)		
Stage I	84 (82.4%)	54 (69.2%)
Stage II	6 (5.9%)	9 (11.5%)
Stage III	6 (5.9%)	10 (12.8%)
Stage IV	6 (5.9%)	5 (6.4%)
Tumor location, *n* (%)		
Body and tail	53 (52.0%)	29 (37.2%)
Body and head	1 (1.0%)	-
Head and neck	44 (43.1%)	45 (57.7%)
Uncinate process	4 (3.9%)	-
AoV	-	1 (1.3%)
N/A	-	3 (3.8%)

AoV, ampulla of Vater; N/A, not available.

**Table 2 cancers-17-02988-t002:** Identified metabolites and their distinct metabolomic features in PC human and KPC mouse plasma compared to their matched controls.

No	Metabolite Identification	Pattern (↑/↓)	Human Plasma			Mouse Plasma		
			*m*/*z*	rt (min)	AUC	*m*/*z*	rt (min)	AUC
1	Acetylcarnitine	↓	204.12197	1.161	0.7956	204.1219	1.138	0.9286
2	Mystamide	↑	228.2311	3.756	0.6715	228.23091	3.75	0.9578
3	Palmitamide	↑	256.2623	4.258	0.7247	256.26229	4.25	0.7857
4	Palmitoylcarnitine	↓	400.34047	5.558	0.6103	400.34014	5.514	0.9058
5	LysoPC(18:3)	↓	518.32194	3.596	0.7287	518.32159	3.605	1.0000
6	LysoPC(18:2)	↓	520.3374	3.863	0.6689	520.33689	3.844	0.7597
7	LysoPC(18:1)	↓	522.35298	4.153	0.6081	522.35272	4.147	0.8474
8	LysoPC(22:6)	↓	568.3375	3.8	0.6431	568.33677	3.785	0.7890

↑: PC (or KPC) > NM, ↓: PC (or KPC) < NM (overall pattern), rt represents retention time.

**Table 3 cancers-17-02988-t003:** CA19-9 and 20 lipids with AUC ≥ 0.75 simultaneously in sets A, B, and C.

Lipid	MRM Transition		rt (min)	AUC		
	Q1	Q3		Set A	Set B	Set C
PE(20:4/20:4)	788.5	647.5	2.32	0.8702	0.8922	0.8805
LysoPC(18:2)	520.3	184.1	0.62	0.7845	0.9313	0.8680
PC(P-18:0/22:6)	818.3	184.1	4.76	0.8697	0.8544	0.8603
C24 Ceramide	650.5	632.5	10.50	0.8024	0.9005	0.8584
PE(P-18:0/22:6)	776.7	385.3	4.64	0.8841	0.9253	0.8537
C10-carnitine	316.4	84.9	4.16	0.8264	0.8564	0.8462
PC(18:0/22:6)	834.6	184.1	4.14	0.8335	0.8444	0.8437
PE(P-16:0/20:4) ^a^	724.5	361.2	3.42	0.7809	0.8558	0.8252
PC(16:1/16:1)	730.5	184.1	2.22	0.7623	0.8483	0.8212
LysoPC(18:3) ^a^	518.3	184.0	0.54	0.8052	0.8858	0.8188
LysoPC(22:6) ^a^	568.3	184.0	0.58	0.8599	0.9152	0.8092
PC(16:0/22:6)	806.6	184.1	2.88	0.8632	0.8523	0.8005
PE(P-18:0/18:2) ^a^	728.6	337.3	5.20	0.8175	0.9000	0.8000
PE(P-18:0/20:4)	752.7	361.1	4.94	0.7914	0.9062	0.7989
LysoPC(20:2) ^a^	548.4	184.0	0.71	0.7873	0.8941	0.7980
PC(18:1/18:1)	786.6	184.1	4.38	0.7518	0.9159	0.7910
PC(14:0/14:0)	678.5	184.1	1.94	0.7504	0.7865	0.7793
PC(18:0/18:0)	790.6	184.1	7.91	0.8150	0.9030	0.7739
PC(16:0/18:2)	758.6	184.1	3.20	0.7531	0.8813	0.7610
LysoPC(20:3) ^a^	546.4	184.0	0.64	0.7539	0.8567	0.7563
CA19-9				0.7571	0.7356	0.7489

^a^ relatively quantified.

## Data Availability

The datasets generated and analyzed in this study are publicly available at the Metabolomics Workbench. Details of study IDs, data types, sample species, and corresponding figures/tables are summarized in Table A3.

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
