# Peer review of "Diagnostic Biomarker Candidates Proposed Using Targeted Lipid Metabolomics Analysis of the Plasma of Patients with PDAC"

_cancers, 2025, doi:10.3390/cancers17182988_

Round 1
Reviewer 1 Report
Comments and Suggestions for Authors
This study presents a timely investigation into lipid-based biomarkers for the early detection of pancreatic ductal adenocarcinoma (PDAC) using targeted metabolomics. The identification of a phospholipid panel showing promising discriminatory power, particularly when combined with CA19-9, addresses a critical clinical need. However, the poor expression in the manuscript significantly hinders comprehension, with the following issues requiring improvement:
- The abstract could be further improved, especially the conclusion section.
- Line 118: At which stage of lipid synthesis or degradation does LysoPC act? How does its increase or decrease affect the disease? Please elaborate on why LysoPC was ultimately selected as a diagnostic biomarker.
- The level of detail in the section describing PDAC lipid metabolic reprogramming (Lines 92-123) is excessive and leads to redundancy. Use a figure to illustrate key pathways instead.
- Alterations in lipid metabolism are observed in many diseases, not just PDAC. How can these changes be specifically distinguished for PDAC?
- The resolution of Figure 2 is insufficient for publication, and the label "ROC curve of mouse samples" is currently invisible; thus, please export and provide a new high-resolution version.
- The font sizes used within the tables are inconsistent. Please revise all tables to use a uniform, standard font size throughout.
- The significant decrease in AUC for multiple lipids in Sets B and C necessitates thorough explanation; please elaborate on the mentioned "batch effects" and specifically discuss their impact on your conclusions.
- The current Reference list contains only 15 items, which is insufficient for the scope of this work. Please add at least 10 relevant, high-quality references.
- Please carefully proofread the manuscript and correct minor errors, such as Line 85 "malignancies3", Line 314 "in references and 14".
- To strengthen the Discussion section, it would be beneficial to incorporate cross-study comparisons of the investigated biomarkers with findings from the cited literature. Additionally, please critically integrate recent publications, assess the current approach's strengths and limitations, and outline future research trajectories.
- Please revise conclusions to provide a more detailed and specific summary of the main outcomes, their significance, and how they address the stated scientific problem. Ensure it accurately reflects the work done and its implications.
- To ensure optimal quality, the text should be polished by a native English speaker or a language professional.
Author Response
Reviewer 1
This study presents a timely investigation into lipid-based biomarkers for the early detection of pancreatic ductal adenocarcinoma (PDAC) using targeted metabolomics. The identification of a phospholipid panel showing promising discriminatory power, particularly when combined with CA19-9, addresses a critical clinical need. However, the poor expression in the manuscript significantly hinders comprehension, with the following issues requiring improvement:
- The abstract could be further improved, especially the conclusion section.
Ans. Thank you. We changed the conclusion as follows,
“Conclusion: We identified 18 candidate fatty acid metabolites that could serve as biological markers in the serum lipid fractions of pancreatic cancer patients and confirmed that all of them were decreased in patients. Additionally, we developed an algorithm utilizing these markers, which demonstrated a 25% increase in discriminatory power compared to the AUC value of CA19-9, an FDA-approved biomarker for pancreatic cancer. In summary, we identified candidate metabolites and algorithms that could serve as biomarkers in the lipid fractions of plasma from patients with pancreatic cancer. “
- Line 118: At which stage of lipid synthesis or degradation does LysoPC act? How does its increase or decrease affect the disease? Please elaborate on why LysoPC was ultimately selected as a diagnostic biomarker.
Ans. Thank you. Whether LysoPC is used for FAO in cancer metabolism requires experimental confirmation, therefore, its significance cannot be explained in this paper. However, I added the following statement to the introduction.
“LysoPC, or lysophosphatidylcholine, is a type of lipid molecule derived from phosphatidylcholine by removing one fatty acid chain. It serves as a key intermediate in lipid metabolism, contributing to the structure of cell membranes and the formation of various other lipids. LysoPC is produced during the breakdown of phosphatidylcholine, a major component of cell membranes. The enzyme phospholipase A2 cleaves a fatty acid from phosphatidylcholine to produce lysoPC [1]. Its role in cancer metabolism is not well understood. However, it can exist as phosphatidylcholine in cell membranes, be degraded and circulate in the blood, or be supplied after digestion in the gut. From an energy metabolism perspective, LysoPC can be further broken down in peroxisomes and metabolized into mid-chain/short-chain acyl-carnitine, which can then be utilized for fatty acid oxidation (FAO) in mitochondria[2]. As a result, its levels are typically lower in cancer patients compared to normal plasma. Therefore, if tumor burden decreases after surgery, these levels can return to normal.”
- The level of detail in the section describing PDAC lipid metabolic reprogramming (Lines 92-123) is excessive and leads to redundancy. Use a figure to illustrate key pathways instead.
Ans. Thank you. We added more information at line 131,
“Cancer cells can use fat or fatty acids in the blood as an energy source. The reason is that there are 50 ~ 200 µM for acetate [3], free fatty acids (100 to 400 µM) [4], and cholesterol (5.2~6.1 mM), LDL (2.6~3.3 mM), HDL (1.0~1.5 mM), and triglycerides (1.7~2.2 mM) in the human blood [5]. Therefore, considering the amounts of various fuel sources for ATP production, including lipids and free fatty acids, cancer cells depend on FAO for ATP production. Furthermore, a recent review has summarized the mechanism by which these lipids and fatty acids are absorbed by cancer cells through various pathways and used in energy metabolism through fatty acid oxidation [2].”
- Alterations in lipid metabolism are observed in many diseases, not just PDAC. How can these changes be specifically distinguished for PDAC?
Ans. That's a good question. However, it's still too early to answer it. No clinically proven biomarkers have been identified for pancreatic cancer, and there is no comparative data from other cancer types. Therefore, the concern about whether we can quickly find differences between cancer types is something that must be addressed later. The human body is a system biology. Each organ performs specific functions, and the metabolism specific to those functions may vary. Elevated AST levels in the blood indicate liver issues. Similarly, different cancer types have distinct metabolic characteristics, and the levels of metabolites change according to these specific features. Fatty acids are one such factor, and analyses to identify differences between cancer types are currently underway.
- The resolution of Figure 2 is insufficient for publication, and the label "ROC curve of mouse samples" is currently invisible; thus, please export and provide a new high-resolution version.
Ans. We apologize for the poor quality of the Figures. We provide a new high-resolution version with a bigger font.
The ROC curve of mouse plasma samples in Figure 2B (4) consists of a vertical line on the left and a horizontal line at the top, since the AUC is 1. These two-line segments are shown as thick lines like Figure 2B (2). We used the AUC = 1.000 value and thick lines in the figure to make the shape of the ROC curve clear.
- The font sizes used within the tables are inconsistent. Please revise all tables to use a uniform, standard font size throughout.
Ans. We apologize for the poor quality of the figures. We are providing a new high-resolution version with larger font.
- The significant decrease in AUC for multiple lipids in Sets B and C necessitates thorough explanation; please elaborate on the mentioned "batch effects" and specifically discuss their impact on your conclusions.
Ans. In 3.3 section, the following sentence was added to explain how we attempted to mitigate batch effects:
The AUCs of sets A, B, and C were compared to exclude lipids with clear batch effects when building statistical models.
- The current Reference list contains only 15 items, which is insufficient for the scope of this work. Please add at least 10 relevant, high-quality references.
Ans. Thanks. We have included more references.
- Please carefully proofread the manuscript and correct minor errors, such as Line 85 "malignancies3", Line 314 "in references and 14".
Ans. Thank you. We fixed the typographical errors and made sure the reference citation numbers are correct. The paragraph with "in references and 14" was revised as follows:
“Lysophosphatidic acid (LPA), a phospholipid, has been used as a biomarker previously [6, 7]; however, it was not available on our phospholipids platform. Previous studies [8, 9] have also discussed ceramide, sphingosine-1-phosphate (S1P), S1P/ceramide, or ceramide/S1P; the C18:1 ceramide was treated as a biomarker by Rixe et al. [8], whereas a specific ceramide was not specified by Guillermet-Guibert et al. [9] Seven ceramides, including C18:1, were available, but S1P was not included in our sphingolipids platform. C18:1 did not meet our criterion of AUC ≥ 0.75 for the measurements we obtained; hence, it was not used as a biomarker in this study. Wolrab et al. [10], using an AI tool, suggested a few sphingomyelins, ceramides, phosphatidylcholines (PCs), and one lysophosphatidylcholine (lysoPC).”
- To strengthen the Discussion section, it would be beneficial to incorporate cross-study comparisons of the investigated biomarkers with findings from the cited literature. Additionally, please critically integrate recent publications, assess the current approach's strengths and limitations, and outline future research trajectories.
Ans. Thank you for your comment. We have revised the discussion section as follows,
“We recently found that cancer cells depend entirely on FAO for ATP production [2, 11]. Additionally, we demonstrated that inhibiting FAO has significant anticancer effects. Based on this, we hypothesized that analyzing blood lipid fractions could distinguish between healthy individuals and cancer patients. However, because the lipid fraction contains various types of fats, we narrowed our focus by comparing mouse cancer models with human samples and performing targeted analyses on matching lipid groups. Consequently, eight fatty acids were identified as matches, and their IDs are listed in Table 2. Among these, LysoPC(18:3) showed an AUC of 0.728 in humans (Figure 2 (B)-(2)) and 1.0 in mice (Figure 2 (B)-(4)). Examining the matched lipid groups between cancer patients and the mouse cancer model in Table 2 reveals three categories: acylcarnitine, fatty acid amide, and phospholipid groups. We also added one more group—sphingolipids—for analysis. We performed targeted lipidomics on human sets A/B/C using these groups. When filtering for AUC values above 0.75, 20 fatty acids were identified (Table 3). Excluding two ceramides, all 18 are from the phospholipid group. Using this data, we analyzed the pattern of LR model AUC changes as more phospholipids were added across two analysis sets (Figure 4). The training set was used to build the model, and the validation set was used to evaluate the model's results. Both sets were selected arbitrarily. Results indicated that after adding 11 phospholipids, the AUC curves of both LR models flattened, regardless of whether CA19-9 was included. The AUC of CA19-9, currently FDA-approved as a biomarker, was 0.7571 in set A, 0.7356 in set B, and 0.7489 in set C. The AUC of the validated model with 11 phospholipids was 0.9207. The combined model achieved a separation performance with an AUC of 0.9427. These findings strongly suggest that measuring specific fatty acids in plasma could effectively determine the progression status of cancer.
Some of the lipids or fatty acids identified as biomarker candidates in this study had been reported previously. Lysophosphatidic acid (LPA), a phospholipid, has been used as a biomarker previously 19, 20; however, it was not available on our phospholipids platform. Previous studies 21, 22 have also discussed ceramide, sphingosine-1-phosphate (S1P), S1P/ceramide, or ceramide/S1P; the C18:1 ceramide was considered as a biomarker by Rixe et al. 21, whereas Guillermet-Guibert et al. did not specify a particular ceramide 22. Seven ceramides, including C18:1, were available; however, S1P was not part of our sphingolipid platform. C18:1 did not meet our criterion of AUC ≥ 0.75 for the measurements we obtained; therefore, it was not used as a biomarker in this study. Wolrab et al suggested several sphingomyelins, ceramides, phosphatidylcholines (PCs), and one lysophosphatidylcholine (lysoPC) 12. Specific PCs and lysoPC species can serve as biomarkers for drug-induced lung disease 23, lung cancer 24, and Alzheimer's Disease 25. Multiple PC species exhibited dysregulated levels in non-small cell lung cancer, particularly increased saturated/monounsaturated forms (e.g., PC (15:0/18:1), PC (18:0/16:0)), and decreased polyunsaturated forms 26. Reduced levels of specific lysoPCs, such as lysoPC(16:0) and lysoPC(18:0), are associated with lung cancer and can help distinguish early-stage lung cancer from other lung diseases 24. Among them, lysoPC(18:2) was also identified in our study. Additionally, PC(32:0) and PC(O-38:5) may correspond to PC(16:0/16:0) and PC(P-18:0/20:4) on our platform for phospholipids. Only lysoPC(18:2), with an AUC of ≥ 0.75, was used as one of the phospholipid biomarkers in this study.”
- Please revise conclusions to provide a more detailed and specific summary of the main outcomes, their significance, and how they address the stated scientific problem. Ensure it accurately reflects the work done and its implications.
Ans. Thank you. We revised the conclusion section as follows;
“Recently, we discovered that cancer cells are completely dependent on fatty acids for ATP production [2, 11]. Therefore, we analyzed the diagnostic potential of specific fatty acid levels in blood by investigating how they respond to tumor growth through animal experiments and patient blood tests. We identified 18 candidate fatty acid metabolites that could serve as biomarkers in the serum lipid fractions of pancreatic cancer patients, all of which were found to be reduced in these patients. Furthermore, we developed an algorithm utilizing these markers and demonstrated 25% improved discriminatory power compared to the AUC of CA19-9. Combining the AUC of CA19-9 with this algorithm further improved discriminatory power by 2.38%. The high AUC values of the model combining the biomarkers identified in this study with CA19-9 suggest that these markers have potential as novel metabolic markers for pancreatic cancer. Therefore, we identified candidate metabolites and algorithms that could serve as biomarkers in the lipid fractions of plasma from patients with pancreatic cancer. More validation sets and multicenter analyses are needed to determine clinically practical implications.
References
- Sato, H.; Taketomi, Y.; Murakami, M., Metabolic regulation by secreted phospholipase A(2). Inflamm Regen 2016, 36, 7.
- Kim, S. Y., Fatty acid addiction in cancer: A high-fat diet directly promotes tumor growth through fatty acid oxidation. Biochim Biophys Acta Rev Cancer 2025, 1880, (5), 189428.
- Pomare, E. W.; Branch, W. J.; Cummings, J. H., Carbohydrate fermentation in the human colon and its relation to acetate concentrations in venous blood. J Clin Invest 1985, 75, (5), 1448-54.
- Schwenk, R. W.; Holloway, G. P.; Luiken, J. J.; Bonen, A.; Glatz, J. F., Fatty acid transport across the cell membrane: regulation by fatty acid transporters. Prostaglandins Leukot Essent Fatty Acids 2010, 82, (4-6), 149-54.
- Lust, C. A. C.; Bi, X.; Henry, C. J.; Ma, D. W. L., Development of Fatty Acid Reference Ranges and Relationship with Lipid Biomarkers in Middle-Aged Healthy Singaporean Men and Women. Nutrients 2021, 13, (2).
- Deken, M. A.; Niewola-Staszkowska, K.; Peyruchaud, O.; Mikulčić, N.; Antolić, M.; Shah, P.; Cheasty, A.; Tagliavini, A.; Nizzardo, A.; Pergher, M.; Ziviani, L.; Milleri, S.; Pickering, C.; Lahn, M.; van der Veen, L.; Di Conza, G.; Johnson, Z., Characterization and translational development of IOA-289, a novel autotaxin inhibitor for the treatment of solid tumors. Immuno-Oncology and Technology 2023, 18.
- Komachi, M.; Sato, K.; Tobo, M.; Mogi, C.; Yamada, T.; Ohta, H.; Tomura, H.; Kimura, T.; Im, D. S.; Yanagida, K.; Ishii, S.; Takeyoshi, I.; Okajima, F., Orally active lysophosphatidic acid receptor antagonist attenuates pancreatic cancer invasion and metastasis in vivo. Cancer Sci 2012, 103, (6), 1099-104.
- Rixe, O.; Villano, J. L.; Wesolowski, R.; Noonan, A. M.; Puduvalli, V. K.; Wise-Draper, T. M.; Curry, R., 3rd; Yilmaz, E.; Cruze, C.; Ogretmen, B.; Tapolsky, G.; Takigiku, R., A First-in-Human Phase I Study of BXQ-350, a First-in-Class Sphingolipid Metabolism Regulator, in Patients with Advanced/Recurrent Solid Tumors or High-Grade Gliomas. Clin Cancer Res 2024, 30, (22), 5053-5060.
- Guillermet-Guibert, J.; Davenne, L.; Pchejetski, D.; Saint-Laurent, N.; Brizuela, L.; Guilbeau-Frugier, C.; Delisle, M. B.; Cuvillier, O.; Susini, C.; Bousquet, C., Targeting the sphingolipid metabolism to defeat pancreatic cancer cell resistance to the chemotherapeutic gemcitabine drug. Mol Cancer Ther 2009, 8, (4), 809-20.
- Wolrab, D.; Jirasko, R.; Cifkova, E.; Horing, M.; Mei, D.; Chocholouskova, M.; Peterka, O.; Idkowiak, J.; Hrnciarova, T.; Kuchar, L.; Ahrends, R.; Brumarova, R.; Friedecky, D.; Vivo-Truyols, G.; Skrha, P.; Skrha, J.; Kucera, R.; Melichar, B.; Liebisch, G.; Burkhardt, R.; Wenk, M. R.; Cazenave-Gassiot, A.; Karasek, P.; Novotny, I.; Greplova, K.; Hrstka, R.; Holcapek, M., Lipidomic profiling of human serum enables detection of pancreatic cancer. Nat Commun 2022, 13, (1), 124.
- Woo, S. M.; Lee, H.; Kang, J. H.; Kang, M.; Choi, W.; Sim, S. H.; Chun, J. W.; Han, N.; Kim, K. H.; Ham, W.; Hong, W.; Kim, C.; Park, J. H.; Han, D.; Yook, J. I.; Lee, W. J.; Kim, S. Y., Loss of SLC25A20 in Pancreatic Adenocarcinoma Reversed the Tumor-Promoting Effects of a High-Fat Diet. Theranostics 2025, 15, (13), 6516-6533.

Reviewer 2 Report
Comments and Suggestions for Authors
The manuscript by Sung-Sik Han et.al. focused on the screening of diagnostic biomarkers in the lipid components of patients with pancreatic ductal adenocarcinoma (PDAC). By screening the common fatty acid types in human and mouse plasma using a-targeted approach, it was possible to determine the targets that could distinguish between normal and cancer patients. Ultimately, a panel of candidate metabolites and algorithm were identified as potential biomers for the lipidomic components of sera from pancreatic cancer patients. The manuscript is logically written. I have three questions:
- Did the author checked the difference of lipidomic candidates between different clinical phases of PDAC patients? Are they decreased gradually accompany the phase increase? What's the function of these candidates in the progression of PDAC?
- How about the correlation of CA19-9 and lipidomic candidates? Why did the author combine them to predict PDAC?
- Figure 1 and A1 are not clear enough.
Author Response
Reviewer 2
The manuscript by Sung-Sik Han et.al. focused on the screening of diagnostic biomarkers in the lipid components of patients with pancreatic ductal adenocarcinoma (PDAC). By screening the common fatty acid types in human and mouse plasma using a-targeted approach, it was possible to determine the targets that could distinguish between normal and cancer patients. Ultimately, a panel of candidate metabolites and algorithm were identified as potential biomers for the lipidomic components of sera from pancreatic cancer patients. The manuscript is logically written. I have three questions:
- Did the author checked the difference of lipidomic candidates between different clinical phases of PDAC patients? Are they decreased gradually accompany the phase increase? What's the function of these candidates in the progression of PDAC?
Ans. Thank you. We have revised abstract, discussion, and conclusion. You can see the meaning.
- How about the correlation of CA19-9 and lipidomic candidates? Why did the author combine them to predict PDAC?
Ans. Since CA19-9 is the only FDA-approved biomarker for PDAC, studies often use it as a benchmark to evaluate the performance of novel biomarkers or assess improvements when used in combination with it. We followed this common research approach.
We added the one sentence on page 11. “Accordingly, all of the lipids were negatively correlated with CA19-9, although the correlations were very weak, with the maximum absolute coefficient being 0.1151 for set C.”
- Figure 1 and A1 are not clear enough.
Ans. We revised the Discussion section as follows to clarify the study strategy.
“We recently discovered that cancer cells depend entirely on FAO for ATP production [2, 11]. Additionally, we demonstrated that inhibiting FAO has significant anticancer effects. Based on this, we hypothesized that analyzing blood lipid fractions could distinguish between healthy individuals and cancer patients. However, because the lipid fraction contains various types of fats, we narrowed our focus by comparing mouse cancer models with human samples and performing targeted analyses on matching lipid groups. Consequently, eight fatty acids were identified as matches, and their IDs are listed in Table 2. Among these, LysoPC(18:3) showed an AUC of 0.728 in humans (Figure 2 (B)-(2)) and 1.0 in mice (Figure 2 (B)-(4)). Examining the matched lipid groups between cancer patients and the mouse cancer model in Table 2 reveals three categories: acylcarnitine, fatty acid amide, and phospholipid groups. We also added one more group—sphingolipids—for analysis. We performed targeted lipidomics on human sets A/B/C using these groups. When filtering for AUC values above 0.75, 20 fatty acids were identified (Table 3). Excluding two ceramides, all 18 are from the phospholipid group. Using this data, we analyzed the pattern of LR model AUC changes as more phospholipids were added across two analysis sets (Figure 4). The training set was used to build the model, and the validation set was used to evaluate the model's results. Both sets were selected arbitrarily. Results indicated that after adding 11 phospholipids, the AUC curves of both LR models flattened, regardless of whether CA19-9 was included. The AUC of CA19-9, currently FDA-approved as a biomarker, was 0.7571 in set A, 0.7356 in set B, and 0.7489 in set C. The AUC of the validated model with 11 phospholipids was 0.9207. The combined model achieved a separation performance with an AUC of 0.9427. These findings strongly suggest that measuring specific fatty acids in plasma could effectively determine the progression status of cancer.
Some of the lipids or fatty acids identified as biomarker candidates in this study had been reported previously. Lysophosphatidic acid (LPA), a phospholipid, has been used as a biomarker previously 19, 20; however, it was not available on our phospholipids platform. Previous studies 21, 22 have also discussed ceramide, sphingosine-1-phosphate (S1P), S1P/ceramide, or ceramide/S1P; the C18:1 ceramide was considered as a biomarker by Rixe et al. 21, whereas Guillermet-Guibert et al. did not specify a particular ceramide 22. Seven ceramides, including C18:1, were available; however, S1P was not part of our sphingolipid platform. C18:1 did not meet our criterion of AUC ≥ 0.75 for the measurements we obtained; therefore, it was not used as a biomarker in this study. Wolrab et al suggested several sphingomyelins, ceramides, phosphatidylcholines (PCs), and one lysophosphatidylcholine (lysoPC) 12. Specific PCs and lysoPC species can serve as biomarkers for drug-induced lung disease 23, lung cancer 24, and Alzheimer's Disease 25. Multiple PC species exhibited dysregulated levels in non-small cell lung cancer, particularly increased saturated/monounsaturated forms (e.g., PC (15:0/18:1), PC (18:0/16:0)), and decreased polyunsaturated forms 26. Reduced levels of specific lysoPCs, such as lysoPC(16:0) and lysoPC(18:0), are associated with lung cancer and can help distinguish early-stage lung cancer from other lung diseases 24. Among them, lysoPC(18:2) was also identified in our study. Additionally, PC(32:0) and PC(O-38:5) may correspond to PC(16:0/16:0) and PC(P-18:0/20:4) on our platform for phospholipids. Only lysoPC(18:2), with an AUC of ≥ 0.75, was used as one of the phospholipid biomarkers in this study.”
References
- Sato, H.; Taketomi, Y.; Murakami, M., Metabolic regulation by secreted phospholipase A(2). Inflamm Regen 2016, 36, 7.
- Kim, S. Y., Fatty acid addiction in cancer: A high-fat diet directly promotes tumor growth through fatty acid oxidation. Biochim Biophys Acta Rev Cancer 2025, 1880, (5), 189428.
- Pomare, E. W.; Branch, W. J.; Cummings, J. H., Carbohydrate fermentation in the human colon and its relation to acetate concentrations in venous blood. J Clin Invest 1985, 75, (5), 1448-54.
- Schwenk, R. W.; Holloway, G. P.; Luiken, J. J.; Bonen, A.; Glatz, J. F., Fatty acid transport across the cell membrane: regulation by fatty acid transporters. Prostaglandins Leukot Essent Fatty Acids 2010, 82, (4-6), 149-54.
- Lust, C. A. C.; Bi, X.; Henry, C. J.; Ma, D. W. L., Development of Fatty Acid Reference Ranges and Relationship with Lipid Biomarkers in Middle-Aged Healthy Singaporean Men and Women. Nutrients 2021, 13, (2).
- Deken, M. A.; Niewola-Staszkowska, K.; Peyruchaud, O.; Mikulčić, N.; Antolić, M.; Shah, P.; Cheasty, A.; Tagliavini, A.; Nizzardo, A.; Pergher, M.; Ziviani, L.; Milleri, S.; Pickering, C.; Lahn, M.; van der Veen, L.; Di Conza, G.; Johnson, Z., Characterization and translational development of IOA-289, a novel autotaxin inhibitor for the treatment of solid tumors. Immuno-Oncology and Technology 2023, 18.
- Komachi, M.; Sato, K.; Tobo, M.; Mogi, C.; Yamada, T.; Ohta, H.; Tomura, H.; Kimura, T.; Im, D. S.; Yanagida, K.; Ishii, S.; Takeyoshi, I.; Okajima, F., Orally active lysophosphatidic acid receptor antagonist attenuates pancreatic cancer invasion and metastasis in vivo. Cancer Sci 2012, 103, (6), 1099-104.
- Rixe, O.; Villano, J. L.; Wesolowski, R.; Noonan, A. M.; Puduvalli, V. K.; Wise-Draper, T. M.; Curry, R., 3rd; Yilmaz, E.; Cruze, C.; Ogretmen, B.; Tapolsky, G.; Takigiku, R., A First-in-Human Phase I Study of BXQ-350, a First-in-Class Sphingolipid Metabolism Regulator, in Patients with Advanced/Recurrent Solid Tumors or High-Grade Gliomas. Clin Cancer Res 2024, 30, (22), 5053-5060.
- Guillermet-Guibert, J.; Davenne, L.; Pchejetski, D.; Saint-Laurent, N.; Brizuela, L.; Guilbeau-Frugier, C.; Delisle, M. B.; Cuvillier, O.; Susini, C.; Bousquet, C., Targeting the sphingolipid metabolism to defeat pancreatic cancer cell resistance to the chemotherapeutic gemcitabine drug. Mol Cancer Ther 2009, 8, (4), 809-20.
- Wolrab, D.; Jirasko, R.; Cifkova, E.; Horing, M.; Mei, D.; Chocholouskova, M.; Peterka, O.; Idkowiak, J.; Hrnciarova, T.; Kuchar, L.; Ahrends, R.; Brumarova, R.; Friedecky, D.; Vivo-Truyols, G.; Skrha, P.; Skrha, J.; Kucera, R.; Melichar, B.; Liebisch, G.; Burkhardt, R.; Wenk, M. R.; Cazenave-Gassiot, A.; Karasek, P.; Novotny, I.; Greplova, K.; Hrstka, R.; Holcapek, M., Lipidomic profiling of human serum enables detection of pancreatic cancer. Nat Commun 2022, 13, (1), 124.
- Woo, S. M.; Lee, H.; Kang, J. H.; Kang, M.; Choi, W.; Sim, S. H.; Chun, J. W.; Han, N.; Kim, K. H.; Ham, W.; Hong, W.; Kim, C.; Park, J. H.; Han, D.; Yook, J. I.; Lee, W. J.; Kim, S. Y., Loss of SLC25A20 in Pancreatic Adenocarcinoma Reversed the Tumor-Promoting Effects of a High-Fat Diet. Theranostics 2025, 15, (13), 6516-6533.

Reviewer 3 Report
Comments and Suggestions for Authors
This article identified candidate metabolites and algorithms that can serve as biomarkers in the lipid fraction of serum from patients with pancreatic cancer. The research perspective of the article is relatively novel, and the data analysis and processing methods are quite advanced. I recommend that this article be accepted after several issues have been resolved.
1.This article only verified the diagnostic performance of lipid markers, but did not attempt to discuss their specific functions in the occurrence and development of PDAC.
2.Only AUC is used in this article to evaluate the model performance. It is recommended to supplement indicators such as sensitivity, specificity, positive predictive value (PPV), and negative predictive value (NPV).
Author Response
Reviewer 3
This article identified candidate metabolites and algorithms that can serve as biomarkers in the lipid fraction of serum from patients with pancreatic cancer. The research perspective of the article is relatively novel, and the data analysis and processing methods are quite advanced. I recommend that this article be accepted after several issues have been resolved.
1.This article only verified the diagnostic performance of lipid markers, but did not attempt to discuss their specific functions in the occurrence and development of PDAC.
Ans. We have revised the discussion, including the meaning of discovery.
We recently discovered that cancer cells depend entirely on FAO for ATP production [1, 2]. Additionally, we demonstrated that inhibiting FAO has significant anticancer effects. Based on this, we hypothesized that analyzing blood lipid fractions could distinguish between healthy individuals and cancer patients. However, because the lipid fraction contains various types of fats, we narrowed our focus by comparing mouse cancer models with human samples and performing targeted analyses on matching lipid groups. Consequently, eight fatty acids were identified as matches, and their IDs are listed in Table 2. Among these, LysoPC(18:3) showed an AUC of 0.728 in humans (Figure 2 (B)-(2)) and 1.0 in mice (Figure 2 (B)-(4)). Examining the matched lipid groups between cancer patients and the mouse cancer model in Table 2 reveals three categories: acylcarnitine, fatty acid amide, and phospholipid groups. We also added one more group—sphingolipids—for analysis. We performed targeted lipidomics on human sets A/B/C using these groups. When filtering for AUC values above 0.75, 20 fatty acids were identified (Table 3). Excluding two ceramides, all 18 are from the phospholipid group. Using this data, we analyzed the pattern of LR model AUC changes as more phospholipids were added across two analysis sets (Figure 4). The training set was used to build the model, and the validation set was used to evaluate the model's results. Both sets were selected arbitrarily. Results indicated that after adding 11 phospholipids, the AUC curves of both LR models flattened, regardless of whether CA19-9 was included. The AUC of CA19-9, currently FDA-approved as a biomarker, was 0.7571 in set A, 0.7356 in set B, and 0.7489 in set C. The AUC of the validated model with 11 phospholipids was 0.9207. The combined model achieved a separation performance with an AUC of 0.9427. These findings strongly suggest that measuring specific fatty acids in plasma could effectively determine the progression status of cancer.
Some of the lipids or fatty acids identified as biomarker candidates in this study had been reported previously. Lysophosphatidic acid (LPA), a phospholipid, has been used as a biomarker previously 19, 20; however, it was not available on our phospholipids platform. Previous studies 21, 22 have also discussed ceramide, sphingosine-1-phosphate (S1P), S1P/ceramide, or ceramide/S1P; the C18:1 ceramide was considered as a biomarker by Rixe et al. 21, whereas Guillermet-Guibert et al. did not specify a particular ceramide 22. Seven ceramides, including C18:1, were available; however, S1P was not part of our sphingolipid platform. C18:1 did not meet our criterion of AUC ≥ 0.75 for the measurements we obtained; therefore, it was not used as a biomarker in this study. Wolrab et al suggested several sphingomyelins, ceramides, phosphatidylcholines (PCs), and one lysophosphatidylcholine (lysoPC) 12. Specific PCs and lysoPC species can serve as biomarkers for drug-induced lung disease 23, lung cancer 24, and Alzheimer's Disease 25. Multiple PC species exhibited dysregulated levels in non-small cell lung cancer, particularly increased saturated/monounsaturated forms (e.g., PC (15:0/18:1), PC (18:0/16:0)), and decreased polyunsaturated forms 26. Reduced levels of specific lysoPCs, such as lysoPC(16:0) and lysoPC(18:0), are associated with lung cancer and can help distinguish early-stage lung cancer from other lung diseases 24. Among them, lysoPC(18:2) was also identified in our study. Additionally, PC(32:0) and PC(O-38:5) may correspond to PC(16:0/16:0) and PC(P-18:0/20:4) on our platform for phospholipids. Only lysoPC(18:2), with an AUC of ≥ 0.75, was used as one of the phospholipid biomarkers in this study.
2.Only AUC is used in this article to evaluate the model performance. It is recommended to supplement indicators such as sensitivity, specificity, positive predictive value (PPV), and negative predictive value (NPV).
Ans. We added the two sentences on page 11:
“For the LR model without CA19-9, the average sensitivity, specificity, PPV (positive predictive value), and NPV (negative predictive value) were 90.74%, 86.22%, 87.90% and 89.42%. They were 90.74%, 88.01%, 89.32% and 89.61% for the LR model with CA19-9.”
In addition, we modified the Results section of Abstract as follows:
“Based on an average AUC for LR models with 11 or more phospholipids, the separation performance between healthy individuals and patients with cancer was 0.9207 (sensitivity, 90.74%; specificity, 86.22%; PPV, 87.90%; NPV, 89.42%), while the addition of CA19-9 to the LR models resulted in a separation performance of 0.9427 (90.74%; 88.01%; 89.32%; 89.61%) for the validation set.”
References
- Kim, S. Y., Fatty acid addiction in cancer: A high-fat diet directly promotes tumor growth through fatty acid oxidation. Biochim Biophys Acta Rev Cancer 2025, 1880, (5), 189428.
- Woo, S. M.; Lee, H.; Kang, J. H.; Kang, M.; Choi, W.; Sim, S. H.; Chun, J. W.; Han, N.; Kim, K. H.; Ham, W.; Hong, W.; Kim, C.; Park, J. H.; Han, D.; Yook, J. I.; Lee, W. J.; Kim, S. Y., Loss of SLC25A20 in Pancreatic Adenocarcinoma Reversed the Tumor-Promoting Effects of a High-Fat Diet. Theranostics 2025, 15, (13), 6516-6533.

Round 2
Reviewer 1 Report
Comments and Suggestions for Authors
The author's revisions are insufficient and require further modifications.
- The authors' statement regarding LysoPC cannot be located in the main text; please confirm that the corresponding content has been added.
- It is recommended that the authors revise the schematic diagram of biomarker screening to be more intuitive and streamline the descriptive text in the main body.
- Figure 2 must be provided in a high-resolution version with significantly enlarged font sizes to ensure all labels, axes, and annotations are legible upon publication.
- Tables 1, 2, and 3 require thorough reformatting to achieve consistency in font size, style, and overall presentation to meet the journal's publication standards.
- The Discussion section should be expanded to include a consideration of potential confounding factors, such as diet, comorbidities, or medications, that may influence the levels of the proposed novel metabolic markers.
- To ensure optimal quality, the text should be polished by a native English speaker or a language professional.
To ensure optimal quality, the text should be polished by a native English speaker or a language professional.
Author Response
- The authors' statement regarding LysoPC cannot be located in the main text; please confirm that the corresponding content has been added.
Ans: In the introduction, we mentioned LysoPC as evidence that fatty acids could be useful biomarkers in pancreatic cancer. However, while other research teams have reported that LysoPC in pancreatic cancer blood has significance as a biomarker, we were unable to find that significance. Since we did not observe this difference in depth through this study, we deleted the LysoPC introduction.
- It is recommended that the authors revise the schematic diagram of biomarker screening to be more intuitive and streamline the descriptive text in the main body.
Ans: We simplified the schematic diagram.
- Figure 2 must be provided in a high-resolution version with significantly enlarged font sizes to ensure all labels, axes, and annotations are legible upon publication.
Ans: We fixed the resolution problem by enlarging the Figures size. The font and size within the image are automatically generated by MS analysis device programs and cannot be manually adjusted.
- Tables 1, 2, and 3 require thorough reformatting to achieve consistency in font size, style, and overall presentation to meet the journal's publication standards.
Ans: We fixed the inconsistancies between Tables.
- The Discussion section should be expanded to include a consideration of potential confounding factors, such as diet, comorbidities, or medications, that may influence the levels of the proposed novel metabolic markers.
Ans: It is currently deemed premature to include potential confounding factors that could influence the values of the new biomarkers in the current discussion session. The current stage involves reporting initial findings and their potential utility, with no evidence to suggest the presence or absence of potential confounding factors. Only after analyzing the results of multiple assessments of their discriminative power and validity across a larger cohort can we determine whether it is necessary to review confounding factors.
- To ensure optimal quality, the text should be polished by a native English speaker or a language professional.
Ans: We have don English editing by MDPI Eng editing service.
